# pH Mapping of Skeletal Muscle by Chemical Exchange Saturation Transfer (CEST) Imaging

**DOI:** 10.3390/cells9122610

**Published:** 2020-12-04

**Authors:** Yu-Wen Chen, Hong-Qing Liu, Qi-Xuan Wu, Yu-Han Huang, Yu-Ying Tung, Ming-Huang Lin, Chia-Huei Lin, Tsai-Chen Chen, Eugene C. Lin, Dennis W. Hwang

**Affiliations:** 1Biomedical Translation Research Center, Academia Sinica, Taipei 115, Taiwan; bcde23400@ibms.sinica.edu.tw (Y.-W.C.); tung@ibms.sinica.edu.tw (Y.-Y.T.); sam320@ibms.sinica.edu.tw (M.-H.L.); rukiya@ibms.sinica.edu.tw (C.-H.L.); 2Institute of Biomedical Sciences, Academia Sinica, Taipei 115, Taiwan; henry84515@gmail.com (H.-Q.L.); qaz950270@gmail.com (Q.-X.W.); vivian1123happy@gmail.com (Y.-H.H.); hazelnut.chen.scu@gmail.com (T.-C.C.); 3Department of Chemistry and Biochemistry, National Chung Cheng University, Chiayi 621, Taiwan; cheel@ccu.edu.tw; 4The Department of Biotechnology, Ming Chuan University, Taoyuan 333, Taiwan; 5The Institute of Biochemistry and Molecular Biology, National Yang-Ming University, Taipei 112, Taiwan

**Keywords:** MRI, CEST, pH, muscle

## Abstract

Magnetic resonance imaging (MRI) is extensively used in clinical and basic biomedical research. However, MRI detection of pH changes still poses a technical challenge. Chemical exchange saturation transfer (CEST) imaging is a possible solution to this problem. Using saturation transfer, alterations in the exchange rates between the solute and water protons because of small pH changes can be detected with greater sensitivity. In this study, we examined a fatigued skeletal muscle model in electrically stimulated mice. The measured CEST signal ratio was between 1.96 ppm and 2.6 ppm in the z-spectrum, and this was associated with pH values based on the ratio between the creatine (Cr) and the phosphocreatine (PCr). The CEST results demonstrated a significant contrast change at the electrical stimulation site. Moreover, the pH value was observed to decrease from 7.23 to 7.15 within 20 h after electrical stimulation. This pH decrease was verified by ^31^P magnetic resonance spectroscopy and behavioral tests, which showed a consistent variation over time.

## 1. Introduction

The intense use of muscles leads to a decrease in power and performance, known as skeletal muscle fatigue [1]. This is a complex physiological phenomenon that manifests as a result of various changes in muscle properties, including changes in the concentrations of extracellular and intracellular ions [2,3]. The accumulation of lactic acid and CO_2_ during muscle activity reduces the cellular pH and subsequently the interstitial pH because of an acid efflux from the muscle cells [4]. Changes in the interstitial pH during muscle activity were suggested to be an important signal in regulating blood flow [4,5,6]. Moreover, a previous study reported a significant decrease in intracellular pH, i.e., from pH = 7.0 to pH = 6.2 during high-frequency skeletal muscle contraction because of muscle fatigue [7]. The relationship between intracellular pH recovery and fatigue-induced alterations suggests that increased H^+^ concentration is a probable cause of fatigue [8,9]. Conventional MRI is capable of providing information on musculoskeletal abnormalities but not on muscle function or fibromyalgia. Therefore, an MRI method to detect and diagnose core muscle dysfunction is required [10].

The biochemistry of skeletal muscles and the chemical conditions within them is important to understand to establish an MRI protocol for muscle diagnosis [10]. In skeletal muscles, creatine kinase (CK), known as creatine phosphokinase or phosphocreatine kinase, is an enzyme reported in various tissues and cell types [11]. CK catalyzes the conversion of creatine (Cr) and adenosine triphosphate (ATP) to phosphocreatine (PCr) and adenosine diphosphate (ADP) [12]. The CK enzyme reaction is reversible, such that the ATP can be produced from PCr and ADP. After muscle contraction, the concentration of lactic acid and Cr is correlated with phosphate content. Lactic acid is produced during muscle exercise and results in the release of H^+^ ions in the muscle cells. Consequently, pH values decrease, and equilibrium concentrations of intracellular metabolites change [12,13,14,15]. The chemical equation for this reaction is given by,
(1)$$\mathrm{PCr}+\mathrm{ADP}+H^{+}↔\mathrm{Cr}+\mathrm{ATP},$$
where the equilibrium constant can be written as follows:(2)$$K=\frac{\left[\mathrm{Cr}\right]\left[\mathrm{ATP}\right]}{\left[\mathrm{PCr}\right]\left[\mathrm{ADP}\right]\left[H^{+}\right]},$$
and in terms of pH, Equation (2) can be rewritten as,
(3)$$\mathrm{pH}=-\log\frac{\left[\mathrm{Cr}\right]\left[\mathrm{ATP}\right]}{\left[\mathrm{PCr}\right]\left[\mathrm{ADP}\right]}+\mathrm{logK}.$$

Rose et al. proposed that intracellular pH can be estimated by the kinase balance [16]. This method was applied by Siesj et al. in studies of rat brain metabolism during arterial hypoxia [17] and hypercapnia [18]. These results indicate that the intracellular pH can be determined by the molar ratio of the PCr, Cr, ATP, and ADP in the total tissue content.

This study aimed to develop an imaging method to quantitatively measure soreness or pain level and detect physiological conditions and chemical properties, i.e., pH values in muscle tissue via MR imaging. However, measuring pH level variation is a technical challenge. The current method used to quantify muscle pH mainly relies on ^31^P magnetic resonance spectroscopy (MRS). However, for ^31^P MRS, the sensitivity of ^31^P by NMR is only 6% of that of ^1^H because of a smaller gyromagnetic ratio; the low sensitivity of this signal causes limited resolution. Localized spectroscopy may be possible for human brain imaging; however, it is challenging to obtain spatially resolved MRS on mouse muscles within a limited time. Chemical exchange saturation transfer (CEST) is a novel sensitivity enhancement technique for nuclear magnetic resonance (NMR) and MRI [19]. In the simplest model, molecules are divided into two pools: a major pool comprising water molecules, and a minor pool comprising metabolite molecules that have protons on their functional groups, such as amines and amides. These metabolites can exchange their proton with the water molecules. Radiofrequency (RF) pulses at a specific transmitter frequency cause the protons on functional groups to be saturated, where saturated means to the point when the rate at which protons absorb the energy and transition to the excited state equals the rate at which protons release energy back to the ground state. The saturated protons exchange with the water protons such that certain protons in the water become saturated. Consequently, the water signal is attenuated in the spectrum or image. The plotting of the water signal as a function of transmitter frequency or chemical shift, with respect to the water frequency, at various frequencies is called the z-spectra.

As mentioned above, the CEST effect comes from the exchange of protons between solute molecules and water molecules. Therefore, hydrogen dissociation from the solute molecules and the proton exchange rate affect the CEST signal, thus making it inherently sensitive to pH. Many previous studies have used amide proton transfer (APT) signal changes in CEST MRI to determine pH in brain and tumor studies [20,21,22]. The exchange rate of amide protons can be accelerated in the presence of hydroxyls [21,23], a process known as base catalysis. Increasing pH results in an enhancement of APT or CEST signals [20,21,22,24]. However, evaluating pH values or obtaining pH-weighted images using only a single measurement of the APT signal is challenging because of the complex composition of biological systems. Alternatively, more accurate pH information could be obtained by considering multiple measurements [25,26,27] or by combining the signal from other saturation offsets [28,29] to remove the confounding factor. Furthermore, different physiological tissues have different types of metabolites and physical and chemical environments. Therefore, pH mapping via MRI requires different methods to respond to other observation objects.

In this study, we developed an imaging method to provide spatial information for pH in muscles. Here, we used CEST MRI to investigate the signal ratio from resonances at 1.96 and 2.6 ppm because of Cr and PCr, respectively [30,31,32,33,34,35,36,37]. This ratio could then be used in conjunction with ^31^P MRS to quantify the intramuscular pH. We then constructed pH mapping over time by testing this new method on a model system in which muscle fatigue was induced by electrical stimulation. Finally, our results were compared with ^31^P MRS and mice behavior tests for validation.

## 2. Materials and Methods

### 2.1. Animal Model

C57BL/6 (~8 weeks old) mice were randomly assigned to experimental (muscle fatigue) (*n* = 6) or control (*n* = 6) groups, both of which were then subjected to CEST imaging and ^31^P MRS. For a von Frey test, we used two legs from the same mouse: one to test muscle fatigue and another as a control (*n* = 5). All procedures involving animals and their care were approved by the Institutional Animal Care and Utilization Committee of Academia Sinica. Mice were housed in standard conditions on a 12-h dark/light cycle with free water access before imaging. During the experiment, body temperature and respiration were maintained at 37 °C by a warm water circulating heating cradle. Mice were anesthetized with isoflurane (1.5%) in oxygen through the process of inducing muscle fatigue and MRI scanning. All mice fully recovered a few minutes after the anesthesia ended.

To induce muscle fatigue, needle electrodes were implanted in the proximal portion of the bilateral gastrocnemius muscle, and mouse legs were straightened during electrical stimulation. Electrical pulses were then applied to the right side of both muscle groups using a modified Burke protocol [38,39,40,41], in which the voltage was set to 5 volts and stimulation was applied by a 500 ms pulse with a 3 s interval. The total stimulation duration was 6 min. To stabilize imaging quality, tungsten wires were used as an electrode to embed the upper and lower ends of the gastrocnemius muscle. The ankle joint’s plantarflexion confirmed the correct electrode placement in the gastrocnemius without activating toe flexors, tibialis anterior muscle, or muscles above the knee. CEST imaging and ^31^P MRS data were recorded at four different time points: before electrical stimulation, 1 h post-stimulation, 3 h post-stimulation, and 20 h post-stimulation. Data from 1-h and 3-h timepoints were used to monitor the instant response to electrical stimulation. Data from the 20 h timepoint was used as reported by a previous behavior study in which fatigue-associated responses to mechanical stimuli increased after 24 h [42].

### 2.2. MRI Measurements

All images were acquired with a Bruker PharmaScan^®^ 7 T MRI Scanner outfitted with a 16-cm-bore (Bruker, Germany). We used a mouse cardiac array coil with a 2 × 2 coil elements topology. *T*_2_-weighted imaging (T2WI) was acquired with a repetition time (TR) of 2500 ms, an echo time (TE) of 33 ms, a field-of-view (FOV) of 25 × 25 mm, a slice thickness of 1 mm, a 256 × 256 acquisition matrix, and an average of 4. CEST imaging was used to acquire anatomical images for pH analysis using the following parameters: TR = 12,263.05 ms, an effective TE of 40.93 ms, a Rapid Acquisition with Relaxation Enhancement (RARE) factor of 8, a FOV of 25 × 25 cm, slice thickness of 1 mm, and an acquisition matrix of 128 × 64. The CEST imaging pulse sequence used a rectangular continuous wave (CW) followed by RARE acquisition. The length of the CW pulse was 3000 ms, and the B1 field was 0.6 μT. The frequency intervals for the z-spectra were: 0.2 ppm between −7 and 1.4 ppm, 0.05 ppm between 1.5 and 2.8 ppm, and 0.2 ppm between 3 and 5 ppm. Baseline frequencies at −333 and 333 ppm were considered as references. To understand the potential effect of temperature during the CEST measurements, a methanol sample was used to test the temperature variation before and after CEST MRI acquisition by measuring the chemical shift between protons on CH and OH groups. The temperature increased 1.1 °C after a one-hour CEST MRI acquisition session.

### 2.3. CEST Imaging

A two-pool chemical exchange model is often used to describe the CEST contrast mechanism, and this provides reasonably accurate quantification of the saturation transfer process. The working principle of CEST includes the usage of a CW RF field to saturate the magnetization of solute protons and subsequently to generate an NMR observable when the protons exchange. Consequently, the signal of the water proton is attenuated. The detailed NMR signal evolution during the CW RF field in the absence of relaxation is given by the following:(4)$$\frac{\partial }{\partial t}\left[\begin{array}{c} m_{x}^{w}({\vec{r}}_{w},t)\\ m_{y}^{w}({\vec{r}}_{w},t)\\ m_{z}^{w}({\vec{r}}_{w},t)\\ m_{x}^{s}({\vec{r}}_{s},t)\\ m_{y}^{s}({\vec{r}}_{s},t)\\ m_{z}^{s}({\vec{r}}_{s},t) \end{array}\right]=\left[\begin{array}{cccccc} -k_{ws} & \delta \omega ({\vec{r}}_{w})/\gamma & 0 & k_{sw} & 0 & 0\\ -\delta \omega ({\vec{r}}_{w})/\gamma & -k_{ws} & -i\omega_{1}/\gamma & 0 & k_{sw} & 0\\ 0 & -i\omega_{1}/\gamma & -k_{ws} & 0 & 0 & k_{sw}\\ k_{ws} & 0 & 0 & -k_{wf} & \delta \omega ({\vec{r}}_{s})/\gamma & 0\\ 0 & k_{ws} & 0 & \delta \omega ({\vec{r}}_{s})/\gamma & -k_{sw} & -i\omega_{1}/\gamma \\ 0 & 0 & k_{ws} & 0 & -i\omega_{1}/\gamma & -k_{sw} \end{array}\right]\left[\begin{array}{c} m_{x}^{w}({\vec{r}}_{w},t)\\ m_{y}^{w}({\vec{r}}_{w},t)\\ m_{z}^{w}({\vec{r}}_{w},t)\\ m_{x}^{s}({\vec{r}}_{s},t)\\ m_{y}^{s}({\vec{r}}_{s},t)\\ m_{z}^{s}({\vec{r}}_{s},t) \end{array}\right]$$
where *s* and *w* denote solute and the water protons, respectively, *k_ws_* is the microscopic rate constant for exchanging the protons from water to the solute, and *k_sw_* is the microscopic rate constant for exchanging protons from the solute to the water. The principle requires a detailed balance of the exchange process such that *P_w_k_ws_* = *P_s_k_sw_* where *P_w_* and *P_s_* denote the populations of spin-bearing molecules of the water and the solute, respectively. The effect of pH is primarily on the exchange rate, i.e., *k_ws_* and *k_sw_*. It affects both the dissociation constant and rate of protons.

The evolution matrix in Equation (4) contains off-diagonal terms and cannot be analytically solved, but can still be investigated by numerical simulation. The magnetization is affected by three off-diagonal terms: off-resonance, RF field, and exchange. Therefore, to enhance the contrast due to the proton exchange, i.e., the effect of the pH variation, the RF field strength must be optimized to gain the proper resolution and signal-to-noise ratio at 1.96 and 2.6 ppm. Several RF strengths were tested, and we used a B_1_ field = 0.6 μT with a 3000 ms duration. In this study, we first used a B_0_ map followed by local second order shimming (voxel size = 4 × 4 × 6 mm) to improve the B_0_ homogeneity.

Moreover, Equation (4) can be used to fit the experimental data for analysis. However, in muscle, the PCr signal appears at 1.96 and 2.6 ppm and hence the sensitivity response of PCr at these two frequencies varies with pH non-linearly, complicating the exchange model. To simplify this analysis, we quantified pH using the signal ratio of 1.96 ppm to 2.6 ppm. To obtain this signal ratio of 1.96 ppm to 2 ppm, we must first subtract the background caused by direct saturation of water molecules and magnetization transfer (MT), and this background signal can be simulated by using Equation (4) (Figure 1). Furthermore, we used two Lorentzian functions to fit the background. Our results show that the two fitting methods were quite consistent. As in previous studies [34,43], Lorentzian difference analysis (LDA) was used to simplify the analysis by fitting a Lorentzian function to the background signal. This was then subtracted from the experimental data. The residual signal was then analyzed using three Lorentzian functions centered at 1.96, 2.6, and 3.5 ppm to obtain the signal ratios at 1.96 and 2.6 ppm. For checking the significance of the difference of pH between different time points and experimental groups, one sample t-TEST was utilized. All image processing and data analyses were then performed in-house using MATLAB^®^ 2020b (The MathWorks, Inc., Natick, MA, USA) and OriginPro^®^ 2020b (OriginLab Corporation, Northamptonshire, MA, USA) software.

### 2.4. ^31^P MRS

Previous studies have shown that the chemical shift difference (Δδ) between PCr and inorganic phosphorus (Pi) is very sensitive to pH [44,45,46]. Moreover, pH can be estimated using the modified Henderson–Hasselbach equation and Δδ using the following equation:(5)$$\mathrm{pH}=6.75+\log\left(\delta -3.27/5.63-\delta \right)$$

All MRS examinations were performed on a Bruker Biospec^®^ spectrometer equipped with a 7 Tesla 21 cm bore horizontal magnet (Bruker, Karlsruhe, Germany) with heteronuclear excitation capability. A ^1^H/^31^P double resonant transmit-receive surface coil was used. For assessing the intramyocellular metabolite chemical shift difference between PCr and Pi, a single pulse-acquire ^31^P-MR spectrum (acquisition delay = 0.4 ms; repetition time = 3 s; bandwidth = 7.8 kHz; 1024 averages in 51 minutes) was acquired at rest. Free induction decay (FID)s were processed with a 10 Hz line-broadening function prior to Fourier transformation.

### 2.5. NMR Measurements of PCr and Cr Phantom Samples

Two phantom samples were prepared for PCr (100 mM) and Cr (100 mM) solutions. pH values were adjusted to 5.6, 6.2, 6.8, 7.4, and 8 by NaOH and HCl. To exclude the effect of concentration on CEST, we prepared PCr (20 mM) solutions and Cr (20 mM) solutions with pH 6.8 and 7.4 for comparison because 20 mM is close to the physiological condition [45]. The signal ratio results of the pH 6.8 and 7.4 samples showed that the concentration effect on the CEST signal ratio with different pH is not significant. Therefore, 100 mM samples were used for all the phantom tests and calibrations.

All ^1^H-NMR experiments were performed on a Bruker Avance III HD 600 MHz WB spectrometer (Karlsruhe, Germany) using a BBFO probe equipped with a z-axis field gradient. The probe temperature was calibrated to 25 °C using methanol. 1H-NMR spectra of phantom samples were acquired using an Ultra-Fast Z-Spectroscopy pulse sequence [47] and maintained at 37 °C. The acquisition parameters used included number of scans = 2, number of dummy scans = 0, receiver gain = 1, spectral width = 10 kHz, acquisition time = 30 ms, relaxation/recovery delay = 20 s, FID data points = 598, and 90° RF pulse duration = 10 µs. FIDs were apodised using an exponential window function prior to Fourier transformation with a line broadening of 10 Hz.

### 2.6. Behavior Test

Mice were placed in a quiet room on a frame with wire mesh for 30 min to acclimatize them to the environment where the von Frey test was conducted during four non-consecutive afternoons. Before paw stimulation, the animals were quiet, without exploratory movements or defecation, and not resting on their paws. On the test day, a 30-min habituation was performed by placing mice on the frame with the wire mesh. Then, a 0.2 g von Frey filament was used by the experimenter to poke the hind paw of the specific mouse. The same paw was poked after an intra-trial interval of 15–20 s. This was repeated 10 times before the other paw underwent the same procedure. The experimental indexes were as follows: mice turn around immediately, jump, and paw lifts.

## 3. Results and Discussion

### 3.1. Subsection

#### 3.1.1. CEST Cr and PCr

According to Equation (3), the correlation between pH and the Cr/PCr ratio can be used to quantify the pH using CEST image analysis. Therefore, the CEST signal response of the two metabolites during the CEST imaging procedure and the relationship between the signal ratio and the pH needed to be calibrated first. These procedures involved understanding the chemical shift and the sensitivity of the signal. Because the CEST signal is derived from the metabolite protons that are exchanged with the surrounding water, the signal intensity is affected by the proton exchange rate. Factors affecting the chemical exchange of the hydrogen include the dissociation ability of the protons on the functional groups of metabolite molecules and the surrounding pH. Therefore, the signal intensity of CEST imaging may change even when the concentration of the metabolite is the same. The measurement of the CEST z-spectra of the Cr and PCr phantoms at different pH values are shown in Figure 2a,b, respectively. In the z-spectra, two peaks can be seen for PCr at 1.96 and 2.6 ppm (Figure 2a), and the Cr peak is at 1.96 ppm with respect to the resonance frequency of the water proton (Figure 2b). These peaks mainly originate from amine groups on PCr and Cr. The CEST signal responses of the two metabolites were very different. The CEST signal for Cr is stronger than that for PCr under the same concentrations and the same pH (Figure 2a,b).

Moreover, the signal intensities changed with different pH values. The integration of the signal area of PCr and Cr at different pH values from Figure 2a,b are shown in Figure 2c. We found that the ratio of the two PCr peaks changed nonlinearly with pH, and that the signal intensity variation with pH was also not linear. As the direct deconvolution of the signals from two metabolites is difficult, the best way to determine pH is to directly correlate the signal ratios at 1.96 and 2.6 ppm with pH. Hence, pH can be written as a function of the signal ratio at 1.96 ppm to that at 2.6 ppm, as given by the following equation:(6)$$\begin{array}{ll} \mathrm{pH} & =f\left(\frac{S_{1.96ppm}}{S_{2.6ppm}}\right)\\ & =f\left(\frac{a(\mathrm{pH})S_{1.96ppm}^{Cr}(\mathrm{pH})+b(\mathrm{pH})S_{1.96ppm}^{PCr}(\mathrm{pH})}{b(\mathrm{pH})S_{2.6ppm}^{PCr}(\mathrm{pH})}\right) \end{array}$$
where $S_{1.96ppm}^{}$ and $S_{2.6ppm}$ are the signals at 1.96 and 2.6 ppm, respectively. $S_{1.96ppm}^{Cr}(\mathrm{pH})$, and $S_{1.96ppm}^{PCr}(\mathrm{pH})$, and $S_{2.6ppm}^{PCr}(\mathrm{pH})$ are the signals for Cr at 1.96 ppm, PCr at 1.96 ppm, and PCr at 2.6 ppm at a specific pH value, respectively. Moreover, *a*(pH) and *b*(pH) are the signal population of Cr and PCr at a specific pH value. To obtain $S_{1.96ppm}^{Cr}(\mathrm{pH})$, $S_{1.96ppm}^{PCr}(\mathrm{pH})$, and $S_{2.6ppm}^{PCr}(\mathrm{pH})$, the data in Figure 2c were fitted by empirical equations for each signal. From these, the signal of Cr at 1.96 ppm, $S_{1.96}^{Cr}(\mathrm{pH})$, and PCr at 2.6 ppm, $S_{2.6}^{PCr}(\mathrm{pH})$, were fitted by the exponential function. PCr at 1.96 ppm, $S_{1.96}^{PCr}(\mathrm{pH})$, was obtained using direct interpolation. To obtain *a*(pH) and *b*(pH), a previous study on the relationship between the Cr/PCr and the pH by metabolomic NMR data was employed [45] and fitted by Equation (3) (Figure 2d). Consequently, Equation (6) was obtained and is shown in Figure 2e. Hence, the signal intensity ratios at 1.96 and 2.6 ppm obtained by CEST imaging were converted to pH values.

#### 3.1.2. Quantification of Cr and PCr by CEST Imaging

After conducting a region of interest (ROI) analysis, the z-spectra of the gastrocnemius muscle before and after electrical stimulation are shown in Figure 3. They reveal an asymmetrical line shape with respect to the frequency resonance of a water proton, denoted as 0 ppm. Because of the B_0_ inhomogeneity in the leg, the central resonance frequency is not uniform throughout the imaging region. Therefore, imaging could only be conducted on one leg, where the magnetic field was relatively homogenous. To correct the frequency in the imaging region, a post-processing method was used, which assumed that the resonance frequency of the water proton was the lowest point of each voxel signal, and therefore was used as the lowest point; this could be used as a reference point whereby the frequency of each voxel was shifted such that their lowest point shared the same frequency. Two peaks could then be seen at 1.96 and 2.6 ppm (Figure 3). Subsequently, data points were subjected to the Lorentzian fitting, as mentioned in Section 2.3. Figure 3b shows the fitting results between 1.75 and 4 ppm using three Lorentzian distributions at 1.96, 2.6, and 3.5 ppm. These three chemical shifts corresponded to different functional groups: the Cr and PCr amine, the PCr amine, and amide, respectively. After comparing the signal intensity ratios of 1.96 ppm to 2.6 ppm to the calibration curve given in Figure 2e, the corresponding pH values could be obtained.

The muscle fatigue induced by electrical stimulation caused acidification in the muscle. By ROI analysis of the gastrocnemius muscle, we were able to track the change in the acid region more clearly. Both the experimental and the control groups are shown in Figure 4a, and the ROI is depicted in Figure 4b. The pH value variation was estimated to lie between 7.15 and 7.25.

Furthermore, the pH of the electrically stimulated region showed a tendency to decrease over time. However, the pH of the control group was steady over time. Moreover, the pH of the control group was ~7.25, which agrees with previous studies that reported that the pH of mice muscles at rest is 7.2 [45,48]. Figure 4c,d show the significant differences between the electrical stimulation and control groups, as well as at different observation times in the electrical stimulation group. The difference in pH values before and one hour after electrical stimulation was not statistically significant. However, this difference becomes significant over longer time scales. When comparing the electrical stimulation group at different observation times, we found no significant difference between the first two time points, i.e., before and one hour after electrical stimulation, but greater, more significant differences were observed at later observation times, i.e., 3 and 20 h.

#### 3.1.3. pH Mapping of In Vivo Muscle Fatigue

The time-dependent pH mapping of the electrically stimulated and control mice legs are presented in Figure 5a,b, respectively. The R^2^ obtained from fitting using three Lorentzian functions on each voxel are also shown. The spatial distribution associated with the contrast in pH mapping showes how the lesion area varied over time. Before electrical stimulation, pH mapping showed that most of the region was estimated to be pH = 7.20–7.25. However, after electrical stimulation, a significant contrast was apparent.

Moreover, a pH of ~7.20 was shown in most regions one hour after electrical stimulation and in most regions at 3 h post-stimulation. Furthermore, pH mapping of the 20 h post-stimulation data showed complete acidification. At this time point, the pH of almost the whole region was close to 7.15. Thus, acidification proceeded by an instant pH change that increased over time. The physiological muscle condition after electrical stimulation is shown by the T2WI of the electrically stimulated mouse leg and pathology results in Figure 5c,d, respectively. The T2WI contrast of the electrically stimulated mouse leg is less at the gastrocnemius muscle and may result from the edema, possibly induced by the electrode, following electrical stimulation. Figure 5d shows the pathology of the muscle sample in the control group and in the experimental group both 1 h and 20 h after stimulation. We observed no significant difference between muscle samples with respect to pathology. These results indicate that muscle fatigue changes the pH of the muscle and not the structure.

#### 3.1.4. Comparison of ^31^P MRS and Behavior Test

To validate the pH mapping results from the CEST imaging, we performed ^31^P MRS analysis (Figure 6a,b). The pH as the function of time following Equation (1) is shown in Figure 6c Because. the ^31^P signal in the animal MRI was weak, the spectrum could only be measured over the entire image range. Therefore, only the average value could be provided by the resulting ^31^P MRS, and it lacked detailed local spatial information. This may consequently cause inaccuracy, but the trend should be roughly comparable. A similar pH variation can be seen as that observed by the pH mapping mentioned above. Over 1–3 h post-stimulation, the data demonstrated that pH changed little and was only slightly different from the ROI analysis conducted on the image. However, data taken 20-h post-stimulation showed an immense change with a pH value ~7.15. This result was close to the observation acquired during the pH mapping.

In addition to the ^31^P MRS data, we used a behavioral test to validate pH mapping. The von Frey test is an extensively used task to measure mechanical nociception responses. There were no notable changes observed in association with the perception of pain stimuli before and after the electrical stimulation for the paws that did not undergo the electrical stimulation (denoted in black in Figure 6d). This result may confirm that there was no problem with the experimental situation and operation. However, a substantial increase was observed for the number of leg lifts associated with the leg that did receive the electrical stimulation (denoted in red in Figure 6d), indicating that it was more sensitive to pain stimuli. To summarize, the operation of the electrical stimulation could change the perception of pain stimuli and make the limb more sensitive. Although the results were not statistically significant, the trend observed was still mostly in line with the time-dependent pH variation in the muscle caused by electrical stimulation. The von Frey test results qualitatively validated the hypothesis that variation in pH is related to the perception of pain. In future studies, functional MRI (fMRI) data can provide more quantitative correlational data between pH mapping and pain perception.

#### 3.1.5. Correlation between Muscle Fatigue and pH Determined by CEST and ^31^P MRS

Figure 7 shows the correlation of pH as determined by CEST and ^31^P MRS. The correlation is almost linear, which validates the method of pH determination by CEST. However, the exact pH values determined by CEST and ^31^P MRS are slightly different. This difference may be caused by the different areas investigated using CEST and ^31^P MRS. As mentioned above, the ^31^P MRS signal represents an average of a whole region covered by the surface coil. Thus, it cannot provide detailed spatial information. However, pH values determined by CEST were obtained from ROI analysis or direct mapping data. These data are more highly localized and offer greater detail. Therefore, in general the pH reading derived from the ^31^P MRS analysis is less sensitive than that from CEST because of the larger average volume.

Moreover, per the muscle pH value and behavior test results, muscle acidity is roughly positively correlated with pain perception. Pathology results demonstrate that this hyperalgesia does not come from muscle damage (Figure 5d). Therefore, muscle pH must influence pain perception and acid metabolism in muscles.

## 4. Conclusions

In this study, we demonstrated the efficacy of pH mapping of skeletal muscle before and after electrical stimulation using CEST MRI data. pH mapping results indicated that muscle tissue was acutely acidified after electric stimulation, but could recover to a similar, pre-stimulus state in ~3 h. The pH mapping at 20-h post-stimulation showed the acidification of the entire leg, as well as a drop in pH value to 7.15. This result was comparable to ^31^P MRS data and behavioral phenotyping tests. Thus, we conclude that CEST MRI is a suitable method for in vivo pH mapping.

## Figures and Tables

**Figure 1 cells-09-02610-f001:**
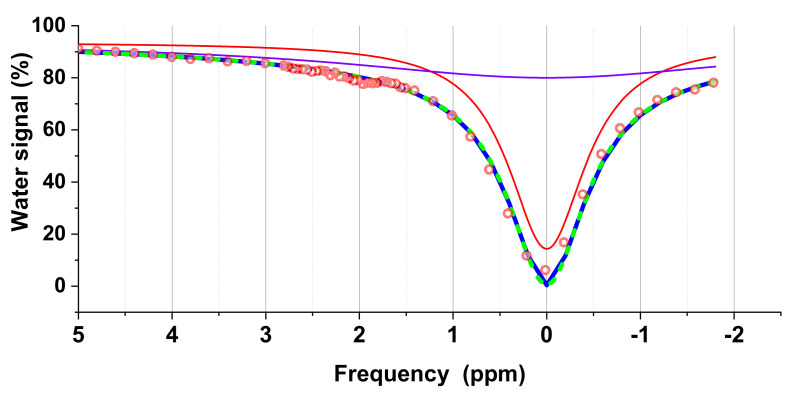
The background curve (blue) is generated by two Lorentzian functions, which include direct saturation of water molecules (red) and the magnetization transfer (MT) (purple), respectively. This background signal also consists of the one simulated by Equation (4) (cyan dash line).

**Figure 2 cells-09-02610-f002:**
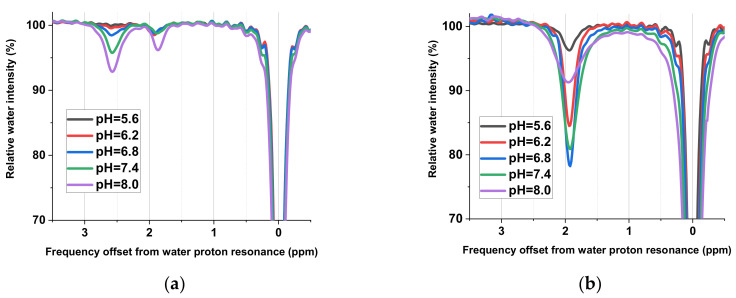
The z-spectra of (**a**) Cr and (**b**) PCr phantoms at various pH values measured by chemical exchange saturation transfer (CEST) NMR spectroscopy. (**c**) The integration (symbol) and fitting (line) results of signals at the peak around 1.96 ppm in (**a**) and 1.96 and 2.6 ppm in (**b**). (**d**) The relationship between the Cr/PCr and the pH was fitted by Equation (3). Data points denoted by ■ were from reference [46]. (**e**) The calibration curve of pH as a function of the ratio of the 1.96 ppm to 2.6 ppm signals, only showing the range used (pH = 7–7.25).

**Figure 3 cells-09-02610-f003:**
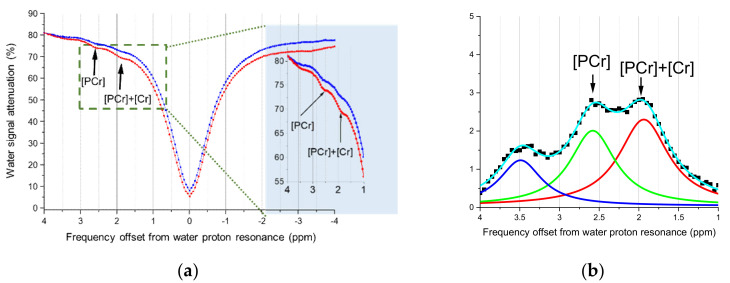
(**a**) The z-spectra of skeletal muscles before (red) and after (blue) electrical stimulation measured by CEST imaging. The box on the right shows a magnified view of the z-spectra between 1 and 4 ppm. (**b**) The original data curve was subtracted from the fitted background curve (the Lorentzian difference analysis (LDA) method). Then, the processed data were deconvoluted by three Lorentzian distributions at 1.96 (red), 2.6 (green), and 3.5 (blue) ppm. Then, the fitting areas were utilized to determine the pH values.

**Figure 4 cells-09-02610-f004:**
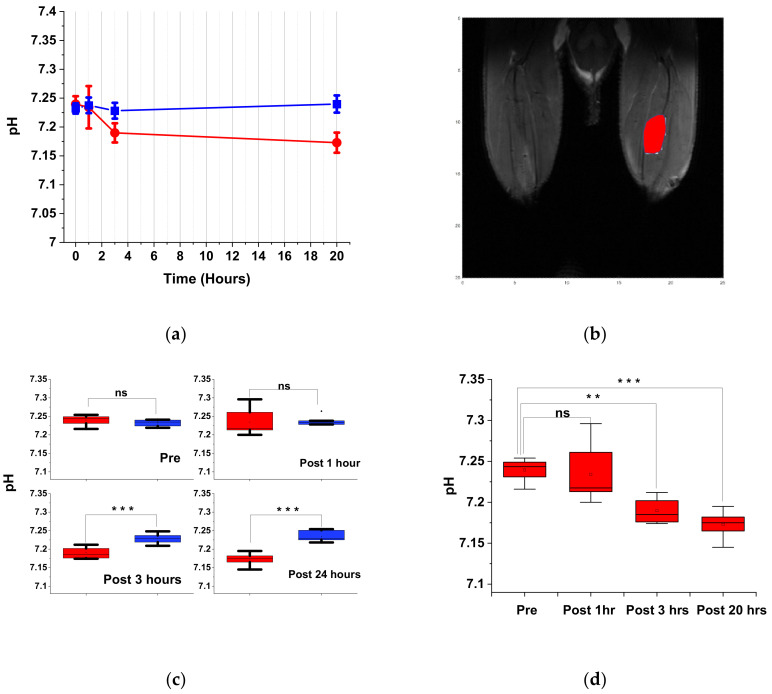
(**a**) Time-dependent pH variation in CEST images by region of interest (ROI) analysis of gastrocnemius muscle region. The ROI is shown in (**b**). The pH of the electrical stimulation group (●) and control group (■) by ROI analysis through time. The 0-h data in the electrical stimulation group refer to the readings taken before electrical stimulation. The deviation between 0-h data in the electrical stimulation and control group may be due to the influence of implanted electrodes (tungsten wires). (**c**) The mean, maximum, and minimum plots of pH of electrical stimulation (green) and control group (red) with different times. A *t*-test was utilized to test statistical significance of the difference in means. The differences between the electrical stimulation and control group at 3-h and 20-h post-stimulation data are significant (*** *p* < 0.001). (**d**) A similar plot showing the pH of the electrical stimulation group at different time points. The *t*-test shows that the differences between the pH readings of pre-stimulation and 3-h and 20-h post-stimulation groups are statistically significant (** *p* < 0.01, *** *p* < 0.001, respectively).

**Figure 5 cells-09-02610-f005:**
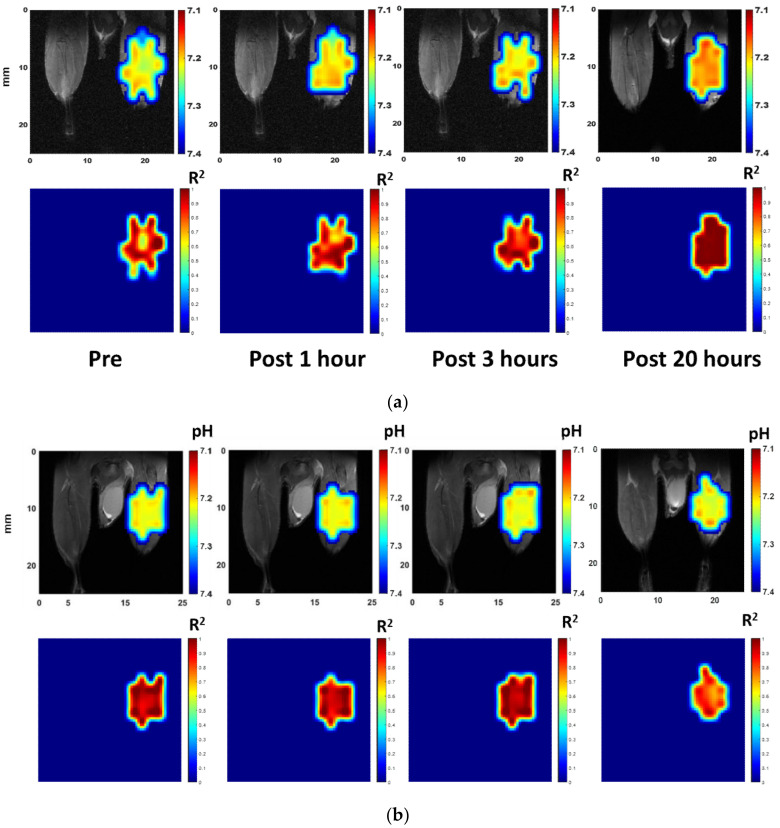
(**a**) The time-dependent pH mapping of the electrically stimulated mouse leg (up) and R^2^ mapping (bottom) to show the coefficient of determination of pH mapping. The pH mapping of the control group and the T2WI of the electrically stimulated mouse leg are presented in (**b**,**c**), respectively, for comparison. The T2WI of the electrically stimulated mouse leg shows only a slightly difference in the gastrocnemius region at 1- and 3-h post-stimulation, and is not distinguishable from the pre-stimulation image after 20 h. (**d**) The pathology of the skeletal muscle of the control group 1 h and 20 h after stimulation.

**Figure 6 cells-09-02610-f006:**
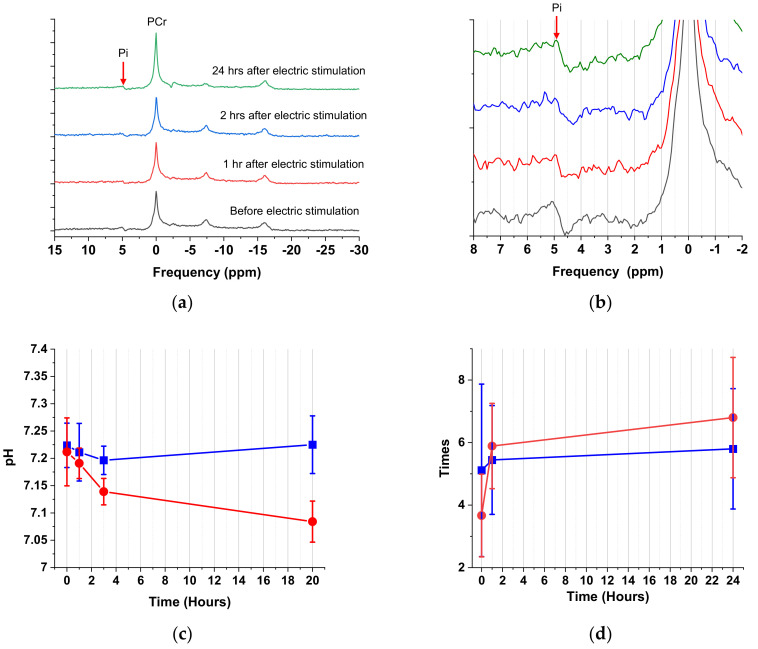
(**a**) ^31^P MRS of the electrical stimulation group. (**b**) A close-up (−2.0–8.0 ppm) of the ^31^P MRS data showing the Pi chemical shift variation over time. The frequency at which Pi moves up-field with time. The chemical shift between PCr and Pi was measured by ^31^P MRS and then using Equation (5). (**c**) The time-dependent pH of the electrical stimulation group (●) and control group (■) using ^31^P MRS analysis. (**d**) The von Frey test results for the paws with (●) and without (■) electrical stimulation.

**Figure 7 cells-09-02610-f007:**
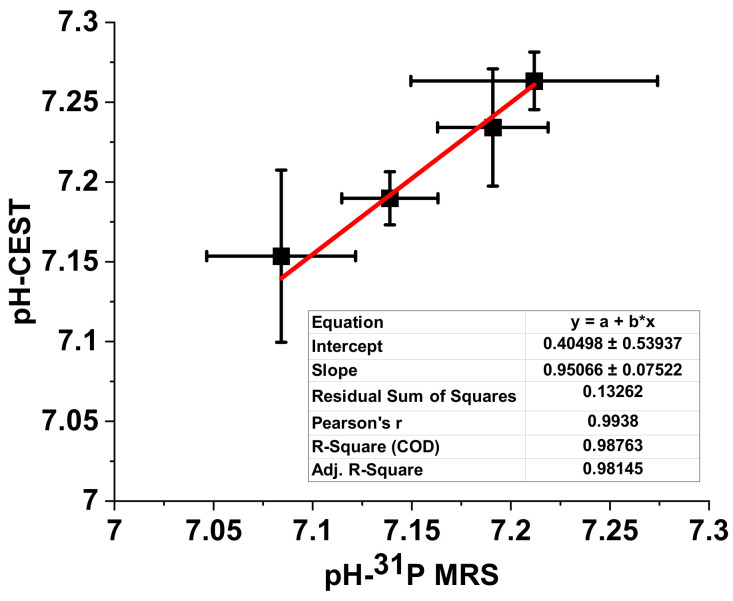
The correlation of pH measurements using ^31^P MRS and CEST methods. The relationship between pH values measured by these two methods is almost linear. The difference may be attributed to differences in the size of the area measured.

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
