# Peer review of "pH Mapping of Skeletal Muscle by Chemical Exchange Saturation Transfer (CEST) Imaging"

_cells, 2020, doi:10.3390/cells9122610_

Round 1

Reviewer 1 Report

This is an interesting study developing and applying CEST MRI for measuring pH change under fatigue-like muscle stimulation longitudinally. However there are many methodological issues and details missing.

Major comments:

  1. English editing by a native speaker is needed.
  2. As there have been studies using similar technique to study muscle pH or creatine, the novelty of the study should be highlighted.
  3. CEST effect is pH dependent disregarding what causes pH change. This property has been used for detecting pH change in stroke and tumors, regardless of the change in Cr/PCr. The relevant literature should be cited and discussed.
  4. Although the study was performed under the Animals Protection Act, whether it was approved by an animal ethic committee needs to be clearly stated.
  5. Animal experiment method is not clear. Was the animal stimulated under anaesthesia or awake, and was that conducted inside MRI? When was MRI/MRS conducted? What is the “Bruker protocol” (line 95)? Is 6-min stimulation enough to lead to fatigue? How was fatigue evaluated or confirmed? Is there a control group as only one (or two?) experimental groups were mentioned? It’s very unclear. How was the animal physiology, particularly the body and muscle temperatures, were monitored and maintained?
  6. More details on CEST scan is needed. Coils used (volume transmit and surface receive?) should be provided. Slice number and thickness are missing. Sweeping frequency, frequency resolution, repetition and total scan time of CEST should be provided. With a 3000ms saturation, the TR shouldn’t be just 2000ms. Please clarify. How B0 field drift over the period of scan was corrected?
  7. In equation (4), how the pH affects the proton signal cannot be seen so it looked like pH independent. Please incorporate pH effect into the model and simulation.
  8. Line 133 mentioned that RF field of CEST needs to be optimized but how it was done is not explained.
  9. No mention of phantom experiment in the Method. What was the concentrate and was that physiological? Although it is easier to measure with Cr and PCr separately, they are not separated in vivo. As Cr and PCr have common exchange site at 1.96ppm, they may interact when they are mixed together. It would be useful to see if the calibration is still the same when Cr and PCr are mixed together. Finally, as CEST effect is also temperature dependent, was phantom measurement conducted at the physiological temperature as the animal?
  10. No explanation on how CEST data was analyzed. Was that fitted to the equation(4)? Was curve fitting (and what order) used? When calculating area, what was the frequency range? Was magnetization transfer ratio calculated (appeared so in Fig.2) and how it was done? Fig.4 showed pH map but it appears to be highly smoothed with resolution very different from that described in the Method section.
  11. No description on the behavior tests.
  12. 1 evaluated CEST at a certain concentration. Since chemical concentration changes the amplitude of CEST effect, how does that compare to the pH effect? Can pH vs concentration change be differentiated?
  13. 1d shows that the calibration with pH only in the range between 7.1 and 7.25. How about below 7.1? It is not sensitive at all to pH > 7.25, which is a critical issue as Fig.3 shows that large pH change happened between 7.2 to 7.4. Since the calibration curve is basically a flat line with pH > 7.25, how was pH=7.3 estimated?
  14. 2: will be useful to show anatomical MRI and corresponding CEST map to show the data quality and where were the CEST z-spectra measured from. Besides show curve fitting in Fig.2b, it will be important to show that result before and after muscle stimulation. Also, the y-axis label in Fig.2b would not be correct if MTR was calculated.
  15. Line 179, what is fitted background curve? Where was that background measured from?
  16. Where does that 3.5ppm signal in Fig.2 come from? Could that be amide proton exchange?
  17. 3: was that measured by CEST or 31P-MRS?
  18. It is strange that muscle acidity got worse over 20hr. Why was the fatigue not recovered? Did the stimulation cause certain damage to the muscle? Please elaborate post-experiment care of the animal. Besides, due to the lack of a control group, it is unclear whether it is the other part of the experimental procedure (ie, long period of MR scans) affect the muscle physiology.

Minor comments:

  1. Description in Line 78 is misleading. Chemical protons don’t exchange with water protons. It is the energy that exchanges.
  2. Wording in Line 80-81 is awkward.
  3. Experiment conducted should be in past tense.
  4. Several abbreviations are used without explanation.
  5. Line 108, B1 field is uT?
  6. Line 124-129, the symbols used in equation should be referred in the same formatting, eg, kws, not just kws.
  7. 1a,b didn’t have legends nor explanation of what each color represents.
  8. Explanation of pH estimation by 31P-MRS should be in the Method section, not Results.

Reviewer 2 Report

The protocol is unclear: 6 muscle fatigue mice and 6 controls (fibromyalgia?) measured by CEST on one MRI system (each mouse all 4 time points?) and the same mice (all 12 mice?) on another MRI system. In the figures only results for fatigue mice seem to be presented. The regions of interest, important in defining the stimulated and surrounding areas, are neither mentioned nor marked in the figures. Results: the CEST acidification in the presumably stmulated area seems nonsignificant (fig.3) and the 0.2 drop in pH cited in the abstract is obtained by MRS only. It is misleading to mention the MRS result without acknowledging that the CEST result is at odds. Must we conclude that either CEST is no good or that the effect of fatigue induction is zero? I would like to see a series of 31P spectra showing the upfield shift of the Pi peak after fatigue in order to enable the reader to judge data quality, also a plot of pH-CEST vs pH-MRS for all individual data points (6x4=24?) to visualise any correlation.

Specific comment: abstract line 20, Cr>Pi

Reviewer 3 Report

Summary: This paper describes the Ph mapping of skeletal muscle using CEST techniques that is verified by 31p MRS. The authors have created a mapping technique using a mouse animal model where exercise was simulated with electrodes in a fibromyalgia and fatigue mouse model. The authors report a drop in Ph from 7.35 to 7.15 within 20 hours.

General comments: The results here are promising. The study is well composed however the paper is missing some important consideration in regards to the execution and assumptions of the results that, as currently presented questions the validity and significance of the results.

Comments:

  • Is there a control group? Authors describe fibromyalgia and fatigue muscle group. Where there any controls/naïve animals run? (section 2.1)
  • The time point extended out 20 hours post treatments. What is the rationale for doing this? Where animals taken out a re-positioned in between time points? Why wasn’t a time point taken before stimulation?
  • Was animal stimulation done with integrated ttl pulses from Bruker ppg (section 2.1 p3 line 96)? Please explain
  • Anesthesia: In brain imaging, isoflurane is sometimes not used because of the effect on the vascular system such as vasoconstriction, reduced body temperature, respiratory effects etc. What (if any) are the concerns of using isoflurane in muscle pH studies where blood supply can a have effect on the outcome? (section 2.1 p3 line 93)?
  • Regardless of Isoflurine, heat is likely to occur during muscle contraction. How is the increased heat in the muscle accounted for since the CEST method used is temperature sensitive? A phantom study can easily be done to show the effect or lack of any effect with temperature.
  • Was the CEST sequence used provided by Bruker sequence library or was a sequence modified with a MT pulse? (section 2.2 p2 line107)
  • The CEST acquisition methods should be clarified and the methods chosen justified. Are the phantom experiments performed in the same fashion as the in vivo experiments? It is not clear reading the method section that phantoms were even performed. More details on the CEST parameters would be beneficial. The authors describe that the RF field strength needs to be optimized to enhance the proton exchange contrast. Although this is true how does this relate to the Ph variation? In Pavuluri et al (https://doi.org/10.1002/ijch.201700075) and Sun et al (1002/cmmi.1680) methods are described involving corrective methods such as PRICEST that accounts for RF spillover effects or using kex as a mapping approach where a function of w1 is used. How does this relate to your study and your study protocol?
  • The authors describe a frequency correction method by the lowest point in the Z-spectra and then fitted by a Lorentzian function. Why was this method used? Why was a method like WASSR not used for B0 correction?
  • Authors state that before electrical stimulation the pH was 7.3. From figure 4a it seems to be to be more yellow/red (~ph=7.2) than blue (7.3)? A roi analysis should be able to determine that. If ROI was used, how was it placed over the muscles?
  • Why does a complete acidification occur at 20 hours when a recovery is seen 3 hours post? How does this compare to naïve animals?
  • Section 3.1.3 has two paragraphs that discuss almost the same things but reference different ph values. Are those for the two animal models used?
  • How is T2w images relevant in the case of Ph mapping? Images of the two animal models and control animals would be of much more relevance to show in figure 4.
  • 31P is a great way to corroborate your results. Why was this done on another magnet and what effects might this have had? What type of coil was used? Was it localized or not, any decoupling, NOE used? How was the data analyzed, what tools? Did you use the same animals or different set of animals? “Raw” 31P spectra should also be presented. How was the long acquisition effecting the results, if any?
  • Same with behavioral test. No description of methods, relevance or adequate results presented.
  • Are the results as would be expected, why? Would you expect to see similar results in humans?
  • Figure 1: Legends are needed on 1 a,b

Reviewer 4 Report

The authors here provided an MRI imaging approach for assessing pH changes in experimental model of muscular fatigue. Although the concept is interesting, the work shows relevant lacks at experimental level, especially in methodology execution.  In addition, the results are weak (probably due to lack of proper control mates) and a proper conclusion is missing. Therefore, it is not now suitable for publication in Cells and this work could be considered after major revision.

Major points:

  1. In the Introduction section the author mention about a decrease of pH values from 7 to 6.2 units during the muscle fatigue. However, the in vivo measurements stands in the range of 7.1-7.4. How then measuring a real pH decrease associated to muscle fatigue?
  2. The pain level concept is reported in the text several times, and it has also tested in the results session. However, a clear association with pH decrease is missing. Please provide some references, or add some information in the Results/Discussion session.
  3. Is the electric stimulation a standard approach for generating animal model of fatigue? How did you extrapolate the parameters used? Please add references in the Methods session.
  4. In the Animal model session, how many animal groups are there? The description is misunderstanding (are muscle fibromyalgia and fatigue muscle models the same group or different? Please clarify the statement, line 91). In addition, a control group (no electrical stimulation) is missing.
  5. Figure 1A: What are the pH values described? Please add in the text and the figure as well.
  6. Line 154. The authors state that “obviously the CEST signal of the Cr is more robust than the PCr”. This statement needs to be explained and rephrased.
  7. Figure 1B/C. How did you perform the calibration? In which solution? Which was the concentration of Cr and PCr? This information is missing but it is important for the reader, and it should be included in the methods session.
  8. Figure 3. The authors said that the pH variation is within the pH 7.1-7.4 units. This is too approximated, since numbers from the graphs are more in the 7.17-7.32 values (mean). Please provide numbers with two decimal units. In such a narrow range it makes huge difference impact.
  9. Imaging of the control leg is missing due to B0 inhomogeneity, therefore imaging of the surrounding tissue is reported. Which is the area determined as “surrounding tissue”? How can the authors clearly define the electric stimulated region and the surrounding one? T2w images (figure 4B) are not able to detect these difference. This is an important issue in the experiment. In fact, figure 3 reveals a drop of pH in the surrounding tissue probably due to the two closed area. The only way to report correct pH data is to repeat the measurement in control animals. I strongly recommend to do this.
  10. The description of Figure 4 is very broad and approximate. In addition, a graph with standard deviation is missing. Please rearrange the text. In addition, images report a drop in pH values until 20h post electric stimulation. I think it would add some additional information to know how the pH changes further, and if this correspond to some physiological changes .
  11. Related to question 10, a demonstration of electric stimulation as model of fatigue is missing. Some changes in physiological markers , or histology , has to be provided in order to confirm the animal model of fatigue is working.
  12. Figure 5A. The comparison with 31P should be performed also in control mice, especially considering that that the spectrum lacks of local information but it has been acquired on the whole image. In addition, the authors claim that big change is reported after 20 hours. However, significancy is not reported.
  13. Figure 5B. The Von Frey test has to be described in the Methods section, or reported with reference. In addition, authors claim a significant difference between stimulated leg and control one. However, statistic is not reported and by observing the error bars it could barely reach the statistical significance. Please rephrased the statement.
  14. Although an interesting concept, a proper correlation graph between pH and pain is missing.  

Round 2

Reviewer 1 Report

Major comments:

  1. In Line 203, “RF field strength needs to be optimized.” However, the next sentence described how B0 field shimming was performed but how RF field was optimized is still not described.

  1. When was 31P MRS conducted? Was it done in the same animal, right after finished CEST imaging? Please clarify in the method section. If different groups of animals were used, it should be clearly stated. If both 31P MRS and CEST were measured from the same animal, then a scatter plot showing individual data point should be provided, instead of just showing the average.

  1. The calibration to pH is a fundamental part of the study. However, it is unclear how the calibration curve in Fig.2d was obtained. Was it calculated based on theoretic estimation or fitting to the empirical data (eg, Fig.2c)? From Fig.2c, the ratio between 1.96ppm vs 2.6ppm CEST signal seems having a very different trend compared to that shown in Fig.2d. If the calibration was derived from certain equations, those details should be provided. Besides, as the result showed that pH estimated by CEST was lower than 7.1 in some animal, the calibration curve should cover the full range of the pH reported.

  1. Line 365-366, “pH mapping of the 3-hours-post-electrical-stimulation showed a recovery from the acidification” in Fig.5, where 1h and 3h post-stimulation showed similar acidity as baseline. However, this is opposite to the results shown in Fig.4, where 3h and 20h post-stimulation both showed acidification. One of them must be wrong.

  1. Several typos and grammar errors.

Minor comments:

  1. Introduction mentioned using APT to measure pH. This study appears to be based on creatine CEST (CrCEST). A bit explanations of APT and CrCEST will be useful.

  1. From Line 209 to 216, an equation was mentioned but its number is missing.

  1. The color coding in Fig.4 should be consistent. If red is used to represent the fatigue group, it should be used in both (a) and (c).

Reviewer 2 Report

After substantial changes, most extraordinary being the switch from rats to mice, the manuscript now seems consistent and logical: acceptable.

Author Response

Thank you.

Reviewer 3 Report

The authors have put in a lot of effort to improve the manuscript. However, it feels rushed and the edits should be proof read again.

  1. Page 3 line 86. The better term would be Magnetic Resonance Spectroscopy (MRS) not NMR even though related.
  2. Page 3 line 101-102. Check grammar.
  3. Page 3 line 111-112. Sentence need revision
  4. Page 4line 149-151. Were the data acquired at 20 hours part of a different study that was also used in this?
  5. Page 4 line 157 and 160, Thickness misspelled
  6. Page 4 line166. Instead of extra, baseline or similar should be used
  7. Page 5 line 210. What equation?
  8. Page6 line 216. What equation?
  9. Page 6 line 246. How are the phantoms relevant if 20 mM solution that according to the authors are close to physiological condition doesn’t show any effect at the physiological pH range?
  10. Page 7 line 273 – 284. Paragraph is confusing and appears repetitive from introduction and methods section.
  11. Page 7 line 286. Resulting misspelled
  12. Page 7 line 289. First work on line misspelled
  13. Page 7 line 295. Should it be figure 2a and b?
  14. Page 7 line 297 – 298. Check for typos
  15. Page 8 line 302. Check for typos
  16. Page 8 line 305 306. Metablomeic NMR?
  17. Page 8 line 327-328. Grammar
  18. Page 8 line 341. Fig 4 correct?
  19. Page 9 line 370 spelling
  20. Page 9 line 273. It is not clear why and how T2W images are used. If the hyper intensities can be caused by edema from the stimulation can’t the CEST data also be affected by it?
  21. Figure 6 a and B. I understand the authors intent with the figure but it is not accurate since signal intensity scale in the x-axis seem to indicate in increased signal for each time point.

Author Response

"Please see the attachment

Reviewer 4 Report

The authors answered the overall questions made by the reviewer. However, minor comments are here presented in order to address this work for publication in Cells.

Revision 2.

  1. Please revise the text. Some English and orthographic typos are present (e.g line 287, 297, 302, 303 àsentences interrupted!, 311, 318, 370…)
  2. Line 346: what do you mean for “monochromatic decrease” ? Please look for other term.
  3. Line 352/356: please specify in the text the time points.
  4. Figure 5 is confusing. In the text (line 359-360), the reference to control groups is missing. For comparison purposes, I think it would be useful to add T2w images also of the electrical stimulated group in figure 5C (is it 5c referring to control group or electrical stimulation? This information is missing).
  5. I am still impressed that a difference of 0.1 in the pH level (from 7.25 to 7.15 after electrical stimulation) could be statistical significant. Which is the error/accuracy of the CEST method used? Which statistical test did you use? Please specify this second statement in the method section.
  6. Line 426: please do not use the word “significant” if the statistic is not significant. In addition, in the electrical stimulated group the increase is more evident because at time 0 the value is lower than for the control group. In addition, is the von Frey study performed on the control group or in the control leg? Please clarify.

Round 3

Reviewer 1 Report

The clarity and quality has greatly improved. All concerns are addressed.

Reviewer 4 Report

The authors provided the modifications required and English language was improved.